# By-Product Revalorization: Cava Lees Can Improve the Fermentation Process and Change the Volatile Profile of Bread

**DOI:** 10.3390/foods11091361

**Published:** 2022-05-07

**Authors:** Alba Martín-Garcia, Montserrat Riu-Aumatell, Elvira López-Tamames

**Affiliations:** 1Departament de Nutrició, Ciències de l’Alimentació i Gastronomia, Facultat de Farmàcia i Ciències de l’Alimentació, Campus de l’Alimentació de Torribera, Universitat de Barcelona, Av. Prat de la Riba 171, 08921 Santa Coloma de Gramenet, Spain; albamartin@ub.edu (A.M.-G.); e.lopez.tamames@ub.edu (E.L.-T.); 2Institut de Recerca en Nutrició i Seguretat Alimentària (INSA·UB), Universitat de Barcelona, Av. Prat de la Riba 171, 08921 Santa Coloma de Gramenet, Spain; 3Xarxa d’Innovació Alimentària de la Generalitat de Catalunya (XIA), C/Baldiri Reixac 4, 08028 Barcelona, Spain

**Keywords:** sourdough bread, volatile compounds, lactic acid bacteria, cava lees, revalorization, wine by-product

## Abstract

Wine lees are a by-product that represents a 25% of the total winery waste. Although lees are rich in antioxidant compounds and dietary fiber, they have no added value and are considered a residue. The aim of this study was to evaluate the effect of Cava lees (0 and 5% *w*/*w*) on microbial populations during sourdough and bread fermentation and the volatile fraction of the final bread. The results showed that 5% Cava lees promoted the growth of both lactic acid bacteria (LAB) and yeast in short fermentations (bread) but did not improve microbial growth in long fermentations (sourdough). Regarding volatile compounds, the addition of Cava lees increased the concentration of volatiles typically found in those products. Also, some compounds reported in sparkling wines were also identified in samples with Cava lees adsorbed on their surface. To sum up, the addition of Cava lees to sourdough and, especially, bread formulation may be a new strategy to revalorize such by-product.

## 1. Introduction

In the EU, around 129 Mt of food waste is generated annually in the food supply chain [1]. It not only has economic repercussions, but it also presents an environmental impact as a consequence of the management and disposal of the food waste [2,3]. The current situation demands for a change from a linear economy to a circular economy where by-products acquire an added value and re-enter the production cycle in order to decrease the environmental impact of industries [4].

It is particularly concerning to the winemaking industry, which includes the production of Cava. Cava is sparkling wine with Protected Denomination of Origin (PDO) that requires a second fermentation in the bottle with a biological ageing process in contact with lees for a minimum of 9 months [5]. Wine elaboration produces approximately 25 kg of waste for every 100 kg of processed grapes. Actually, the main solid residues produced by winemaking are grape pomace (60%), lees (25%) and stalks (15%) [2,3].

Lees are the residue formed during wine fermentation and consist, mainly, of naturally plasmolyzed cells of *Saccharomcyces cerevisiae*, tartaric acid and other adsorbed compounds [6,7]. It is estimated that the production of Cava lees is about 300 tones per year [8]. Those lees are rich in phenolic compounds as well as fiber and proteins from the cell wall of *S. cerevisiae* [2,5,6,7]. Indeed, the use of by-products with high contents of fiber and other bioactive compounds as novel ingredients is being studied to obtain foods with greater nutritional value [9,10,11].

Despite their composition, lees are an undervalued by-product mostly used for the recovery of tartaric acid and distillation to obtain alcohol [12]. Nevertheless, some studies have reported the possibility of revalorizing wine lees [8,13,14,15].

In fact, Hernández-Macias et al. (2021) [6] reported an improvement of growth and survival of lactic acid bacteria (LAB) with the addition of Cava lees in vitro. In addition, our research group recently demonstrated that the addition of Cava lees to sourdough formulation promoted the growth and survival of microorganisms (both LAB and yeast) in spontaneous fermentation [15]. In addition, it has been reported that Cava lees can inhibit the growth of pathogens (*Salmonella* spp. and *L. monocytogenes*), improving the microbiological safety of fermented sausages [6]. Hence, Cava lees might be revalorized as a food ingredient to improve fermented foods like sourdough and bread. Moreover, modifying microbial populations of food fermentation might have an impact on its flavor [16].

Indeed, flavor and especially odor are of great importance for consumer acceptance. It must be taken into account that the addition of by- and co-products to food formulation may change its sensory properties (from texture to aroma or color). For instance, Lafarga et al. (2018) [9] added broccoli co-products (stems and leaves) to wheat bread formulation to obtain functional products with enhanced concentrations of fiber and phenolic compounds. The researchers observed that breads with broccoli presented an increased green hue as well as a higher color intensity in crumb and crust. Nevertheless, when performing sensory tests, the overall acceptance of the breads was not affected by broccoli incorporation [9]. Other by-products, such as cumin and caraway seeds by-products and cocoa dietary fiber have been added to wheat bread formulation to obtain functional products [10,11]. In both studies, researchers examined the effect of those new ingredients on the sensory properties of bread. In both cases, there were no significant differences on the overall acceptance of the fortified breads and controls, even though color, texture and aroma changed.

In that regard, lees are able to adsorb volatile compounds during the biological ageing of sparkling wine [17]. Consequently, incorporating Cava lees to sourdough and bread formulations may add new odors and other compounds to such bakery products. Therefore, the aim of this study was to evaluate the effect of Cava lees on microbial populations during sourdough and bread fermentation as well as the volatile fraction of those breads to revalorize this winery by-product.

## 2. Materials and Methods

### 2.1. Sourdough Formulation and Bread-Making

A commercial wheat flour (Harina de Fuerza Gallo, Comercial Gallo S.A., Barcelona, Spain) with the following composition (% *w*/*w*): carbohydrates 69.0, fat 1.4, fiber 4.2, protein 11.7 and moisture 15.0, was used.

A parallel study was designed in order to compare breads with and without sourdough. Sourdoughs were prepared by mixing 100 g of flour and 100 mL of sterile distilled water (Table 1), without the inoculation of microorganisms and incubated at room temperature for 24 h. Cava lees were lyophilized following the method described by Hernández-Macias, Comas-Basté, et al., 2021 [6]. They were added as a percentage of flour weight at 5% (*w*/*w*) and compared to a control without lees, based on previous in vitro studies [6]. Sourdoughs were propagated by backslopping for 8 days, inoculating an aliquot of the previous dough into a new mixture of flour and water, adding the corresponding lees percentage.

Breads were made with the sourdoughs produced (Table 2). Breads were prepared by mixing flour (500 g), water (285 mL), sourdough (150 g), baker’s yeast (4 g) and salt (10 g). Separately, breads fermented with commercial yeast (Ref.: 36835, Sosa Ingredients S.L., Barcelona, Spain) were also prepared with the following formulation: flour (500 g), water (285 mL), Cava lees (0% and 5% (*w*/*w*)), and salt (10 g). Cava lees were also added as a percentage of flour weight. The dough was manually mixed and kneaded for 10 min. The dough temperature at the end of kneading was between 22 and 24 °C. Once formed, dough was rested for 40 min, after which the dough was knocked back and rested for another 40 min. All bread was given a final proof of 20 min at 30 °C and 80% relative humidity. Following the processing of the dough, breads were baked in a convection-steam oven (Ref.: SA-SC-623, Salva S.L.U., Guipuzkoa, Spain) at 220 °C for 30 min.

### 2.2. Microbial Populations and Fermentation Monitoring

Microbial populations were monitored following the method described by Martín-Garcia et al. (2022) [15]. Briefly, samples of 10 g were added to 90 mL of sterile peptone water (Ref.: 1402, Condalab, Madrid, Spain) and homogenized with a laboratory blender (Stomacher 400 Seward Ltd., Worthing, UK) for 1 min. Sourdough samples were taken daily, while breads were monitored every 30 min. All samples were diluted and plated in MRS (Ref.: 1043, Condalab, Madrid, Spain) to monitor LAB populations and in Saboraud-Chloramphenicol Agar (Ref.: 01-166-500, Scharlab, Barcelona, Spain) for yeasts. Also, pH was monitored in all samples using a pH meter XS PH60 VioLab (XS Instruments, Carpi MO, Italy).

### 2.3. Headspace Solid Phase Microextraction (HS-SPME)

The extraction of volatile compounds was performed using Headspace Solid Phase Microextraction (HS-SPME) as reported by Paraskevopoulou et al. (2012) [18] using a 2 cm long Divinylbenzene/Carboxen/Polydimethylsiloxane (DVB/CAR/PDMS) fiber supplied by Supelco (Bellefonte, PA, USA). Before extraction, the fiber was conditioned according to the manufacturer’s recommendations. All breads were grinded, and samples of 3 g were placed in 20 mL vials. Then, 1 mL of a 20% NaCl solution (pH 3 adjusted with 0.05 M citric acid solution) was added to the vial. After equilibration at 60 °C for 30 min, the fiber was exposed to the sample headspace for 60 min. An internal standard [4-methyl-2-pentanol (CAS: 108-11-2, TCI Ltd., Eschborn, Germany), 100 µg/L] was used (100 µL) for semi-quantification.

### 2.4. Analysis of Volatile Compounds by Gas Chromatography—Mass Spectrometry (GC-MS)

Chromatographic analysis was carried out in a 6890N Network GC system coupled to MS 5973 Network selective detector (Agilent, Palo Alto, CA, USA). Helium was used as a carrier gas. Separations were accomplished in a DB Wax USN 125-7031 column (30 m × 0.25 mm × 0.25 µm) (Agilent, Palo Alto, CA, USA). A splitless injector suitable for SPME was used. After extraction, the fiber was removed from the headspace vial and manually inserted directly into the injection port of the GC. The SPME fiber was thermally desorbed for 4 min at 260 °C.

The initial temperature was held at 40 °C for 5 min and increased at from 40 °C to 190 °C at 3 °C/min and from 190 °C to 220 °C at 10 °C/min which was held for 5 min using splitless injection mode. GC-MS detection was performed in complete scanning mode (SCAN) in the 40–350 amu mass range with two scans per second. Electron impact mass spectra were recorded at an ionization voltage of 70 eV and ion source of 280 °C. Volatile concentrations reported were calculated by dividing the peak area of the compounds of interest by the peak area of the internal standard (normalized area). The relative response factor was considered to be 1. Tentative Identification was performed by comparison of their mass spectra with those of the mass spectra library database Wiley 6.0., and their retention times with those of pure standards when they were available.

### 2.5. Statistical Analysis

The statistical analysis was performed using Prism 9 version 9.1.2 (225) (GraphPad Software, LLC., San Diego, CA, USA) statistical package. The results are reported as the means ± standard error (SE) of triplicates for parametric data. A one-way ANOVA and comparison of the means were conducted using Tukey’s test, with a confidence interval of 95% and significant results with a *p*-value of ≤0.05. Principal component analysis (PCA) was also performed to determine differences between breads.

## 3. Results and Discussion

### 3.1. Microbial Populations and Fermentation Monitoring

A control and a fortified (5% Cava lees) sourdough were prepared to assess the effect of Cava lees on the fermenting microbiota of sourdough. Figure 1 shows the growth kinetics of lactic acid bacteria (LAB) and pH during the 8 days of sourdough propagation. Both types of sourdough (control without lees—SDC; with 5% lees—SD+L) were spontaneously fermented. Although the promoting effect of 5% (*w*/*w*) Cava lees on LAB growth has been reported in vitro [6] and up to a 2% (*w*/*w*) Cava lees in wheat and rye sourdoughs [15], it can be observed that the addition of Cava lees to sourdough formulation did not stimulate LAB growth in this particular food matrix.

The initial pH of sourdoughs with 5% Cava lees (*w*/*w*) was significantly lower (*p* < 0.05). In particular, SDC started the sourdough fermentation with a pH of 5.80 ± 0.01, while SD+L begun with a pH of 5.06 ± 0.07 due to the inherent acidity of Cava lees [8]. During the fermentation and propagation process of sourdough, pH decreased during the first steps of fermentation and then stabilized, obtaining values of 3.60 ± 0.04 (SDC) and 3.70 ± 0.02 (SD+L), similar to those reported in other studies [19,20,21] and in accordance with the pH range of traditional sourdoughs (pH 3.5–4.5) [13]. However, there were no statistically significant differences between sourdoughs. Once sourdoughs were mature (8 days), breads were prepared (Table 2) and baked.

Figure 2 presents the cell density of both LAB and yeast at the end of bread fermentation. When Cava lees were used in the formulation of bread (both fermented with and without sourdough) there was a higher cell count for both yeasts and bacteria. In fact, bread fermented without sourdough presented a difference of 0.7 log10 CFU/mL between the ones with Cava lees (B+L) and the controls (BC). Also, there was a higher cell density in SB+L (5.1 ± 0.2 log10 CFU/mL) in comparison to SBC (4.1 ± 0.1 log10 CFU/mL). Regarding yeasts, the addition of lees to formulation had the same tendency.

Sourdough bread usually presents a pH ranging between 5.0 and 5.5 [22], which was in accordance with the results obtained in this study (Table 3). The addition of Cava lees to both sourdough bread (SB+L) and leavened bread (B+L) resulted in lower pH at the beginning of dough fermentation, and, consequently, also at the end. As previously stated, the difference in pH values between samples with and without lees was probably due to the inherent acidity of Cava lees [8]. In fact, B+L obtained the lowest pH values of all formulated doughs. Actually, the addition of Cava lees to bread formulation (B+L) resulted in the greatest drop of pH during fermentation, which could be related to the higher plate counts of both LAB and yeast (Figure 2).

### 3.2. Analysis of Volatile Compounds

In order to evaluate the effect of Cava lees on the volatile fraction of breads, HS-SPME-GC-MS was performed. A total of 74 volatile compounds were identified (Table 4), including nine acids, 16 alcohols, 11 aldehydes, five ketones, 14 esters and eight terpenes. Bread volatile compounds may result from fermentation, lipid oxidation and Maillard and caramelization reactions [20,23,24,25]. Alcohols, acids, esters, aldehydes and ketones were generated mainly during fermentation while some of them as alcohols, ketones and aldehydes come from lipid oxidation too.

Lastly, Maillard and caramelization reactions originate pyrazines, pyridines, pyrroles, furans, sulfur compounds, aldehydes and ketones [22]. Additionally, the volatile compounds of Cava lees were also analyzed by HS-SPME (Appendix A), since wine lees are able to retain aromatic substances such as esters, aldehydes, norisoprenoids, terpenes and some phenolic compounds [17,26].

In general, B+L had the highest concentration and number of different volatile compounds (*p* < 0.05), especially in acids (660.46 ± 135.75 mg/kg), esters (1804.88 ± 274.08 mg/kg) and terpenes (215.74 ± 27.95 mg/kg). Oppositely, controls (SDC and BC) were richer in alcohols (611.27 ± 80.37 and 488.17 ± 72.70 mg/kg, respectively).

Although it would be expected that sourdough bread had a richer aroma profile [29], in the present study we obtained less abundance of volatile compounds in SB+L and SBC breads. In that regard, sourdough is generally added at less than 50% of the flour content (Table 2) and afterwards there is a baking process, so volatile compounds from sourdough might be diluted in the end product [24].

Acids are a product of the fermentation process and are responsible for the acidification of the dough [24,30]. Nevertheless, organic acid production during sourdough and bread-making depends on several variables, including microbial composition as well as process parameters (dough yield, fermentation time and temperature and NaCl concentration) [16,24,30]. In B+L samples, the dominant acids were dodecanoic acid (240.41 ± 83.22 mg/kg) and octanoic acid (173.53 ± 11.96 mg/kg). In fact, octanoic acid along with decanoic acid, were the major organic acids found in Cava lees (Appendix A). Octanoic acid was also found in SBC and BC samples, and its production is related to yeast [24].

Control breads with and without sourdough (SBC and BC) presented no significant differences in organic acid concentration (*p* < 0.05). Furthermore, SBC and BC breads had the highest concentration of acetic acid (143.83 ± 1.63 and 132.04 ± 0.06 mg/kg, respectively). In fact, acetic acid is one of the main organic acids responsible form microbiological shelf-life extension since it also possesses antiripeness and antimold activity [20,30]. Moreover, acetic acid is thought to inhibit yeast growth [25], which can be related to the lower yeast cell density obtained in SBC and BC breads (Figure 2).

Alcohols are mainly produced during fermentation from flour amino acids via the Ehrlich pathway in yeast cells but may be also a product of lipid oxidation [20]. SBC showed the highest concentration of alcohols (*p* < 0.05). The dominant alcohols in all bread samples were phenethyl alcohol and isoamyl alcohol. Phenethyl alcohol is derived from the fermentation of phenylalanine by yeast, and it has been reported that prolonged fermentations increase its concentration [20,31]. SBC had the highest concentration (253.45 ± 3.78 mg/kg), which can be related to the longer fermentation of sourdough. Isoamyl alcohol is a product of the fermentation of leucine also in the yeast cell [20,24,31] and presented higher values in sourdough samples (with and without Cava lees). In addition, isoamyl alcohol can also be produced by LAB such as *Lactiplantibacillus plantarum* (formerly *Lactobacillus plantarum*) [24,32].

Aldehydes are formed during lipid oxidation and decarboxylation of unsaturated fatty acids as well as from amino acid degradation by the Ehrlich pathway [20,33]. The most prevalent aldehydes found in the studied breads were benzaldehyde, hexanal, (E)-2-nonenal and nonanal which are commonly reported in both sourdough and bread [20,24,33].

The addition of Cava lees increased the concentration of benzaldehyde, especially in yeast leavened bread (B+L, 88.66 ± 11.41 mg/kg). This compound is the result of both fermentative reactions and lipid oxidation and has been found in bread produced with and without sourdough [20,24,34]. Benzaldehyde has been reported to be produced by yeast as well as *L. plantarum* and *L. helveticus* via amino acid (phenylalanine) conversion [24,35]. Moreover, benzaldehyde has also been found in sparkling wines [36,37,38], which might explain the increase in its concentration in breads with Cava lees since sparkling wine lees can retain aldehydes in their surface (Appendix A) [17].

Nonanal, another compound derived from fermentation and lipid oxidation [20] has also been identified in the surface of sparkling wine lees [17]. In this study, SB+L and B+L showed significantly higher amounts of this compound, that was, indeed, also identified in Cava lees surface (Appendix A). In fact, in SBC, nonanal was not detected. It must be taken into account that some heterofermentative LAB strains are able to reduce aldehydes to other compounds, which may explain the lower concentration of certain volatiles in samples with 5% Cava lees [32,35].

Regarding ketones, BC samples presented a greater variety of those compounds. Those volatile compounds are influenced by LAB in dough fermentation, and only certain homofermentative and facultatively heterofermentative bacteria are able to produce them [32]. Acetoin is a distinct aroma in bread produced during fermentation related to consumer acceptance [23]. In this study it was found that the addition of Cava lees increased its production, especially in B+L where it reached similar values to those of sourdough bread.

Esters are characterized by a fruity odor resulting from a direct esterification between ethanol and acetyl co-A derivatives of fatty acids during fermentation mainly due to heterofermentative LAB [32,35,39]. In fact, it has been observed that fermentations with LAB produce more esters than those with yeast [24]. In this study, the addition of 5% Cava lees increased the production of esters, especially in bread samples (B+L) in which 13 esters were identified, which is in accordance with higher LAB populations (Figure 2). Ethyl decanoate was the dominant ester in SB+L (127.37 ± 13.90 mg/kg), while in B+L it was ethyl octanoate (965.31 ± 167.18 mg/kg). As previously mentioned, sparkling wine lees also retain esters such as ethyl hexanoate, ethyl octanoate and isoamyl octanoate [17]. In fact, most of the esters only identified in B+L samples have been reported in sparkling wine [36,37,40] and sparkling wine lees [17], and were also found in the Cava lees analyzed (Appendix A).

Terpenes are generally characterized by a floral odor and commonly found in sparkling wine [37,38]. Furthermore, vitispirane and nerolidol, identified in this study, have also been found in sparkling wine lees [17]. Overall, it was found that the addition of Cava lees increased terpenes concentration and variability in both sourdough bread and, especially, yeast leavened bread.

TDN (1,1,6-trimethyl-1,2-dihydronaphthalene) is a C13-norisoprenoid usually found in sparkling wine [36,37,38]. TDN has been pointed out as an ageing marker in sparkling wine, along with diethyl succinate (ester) and vitispirane (terpene) [37]. It was identified in both SB+L and B+L but not in the respective controls. Those compounds were found by Gallardo-Chacón et al. (2009) [17] in sparkling wine lees surface, as well as in our Cava lees analysis (Appendix A).

Lastly, the results obtained were subjected to a PCA to determine the differences between the breads produced with and without sourdough and Cava lees. Figure 3 shows the result of a previous correlation analysis and Figure 4 presents de PCA biplot obtained. The PC1 and PC2 explain 79.02% of the total variability. The first principal component (PC1) explains a 52.37% of the samples variances while the second one (PC2) explains a 26.64%. Most of the volatile compounds were found in the positive side of PC1, especially esters, acids, and linear aldehydes, whereas branched aldehydes, alcohols and ketones were situated on the negative axis of PC1. On the other hand, the positive axis of PC2 contained a greater number of volatiles, including alcohols, linear aldehydes, ketones, and esters; while branched aldehydes and terpenes were situated in the negative side of PC2.

It can be observed that both controls (SBC and BC) were placed in the same quadrant. In fact, both controls and SB+L were positioned opposite bread with Cava lees (B+L). SBC and BC were characterized by alcohols and ketones, while B+L was described by esters and terpenes. Moreover, a greater quantity of different volatiles was identified in B+L samples.

## 4. Conclusions

There are several factors involved in the development of bread flavor, from microbial activity to aroma precursors in the flour used. Therefore, it is important to determine the volatile fraction of the product to obtain consumers acceptance. Formulation of bread with 5% Cava lees (*w*/*w*) improved microbial growth (both LAB and yeast) in short fermentations, although there were no significant differences in prolonged fermentations (sourdough). Actually, LAB and yeast release aroma compounds as well as aroma precursors (including carbohydrates and amino acids) that can be transformed into the corresponding volatiles. Thus, higher microbial populations obtained with Cava lees might produce a greater concentration of volatile compounds due to LAB and yeast fermentation in dough. In general, the addition of Cava lees to bread increased the concentration of volatiles typically found in bread and sourdough bread. Also, some compounds usually reported in sparkling wines were also identified in samples with Cava lees, supporting the fact that yeast lees adsorb volatile compounds during wine ageing.

Therefore, it can be concluded that Cava lees promote the production of bread volatiles besides contributing with new odors from sparkling wine. Hence, the use of Cava lees as an ingredient in bread fermentation could be a new strategy to revalorize this winery by-product obtaining a new bread product. Also, further studies should focus on the effect of Cava lees on identified microorganisms of sourdough and bread, since the fermenting microbiota can influence bread aroma and flavor.

## Figures and Tables

**Figure 1 foods-11-01361-f001:**
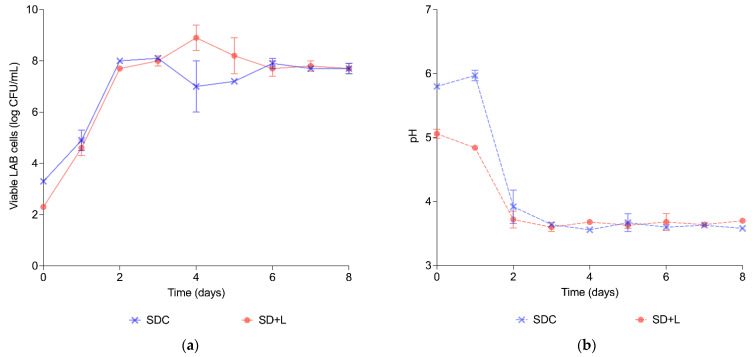
Growth of LAB (**a**) and pH (**b**) in sourdough without lees (SDC) and sourdough with 5% lees (SD+L).

**Figure 2 foods-11-01361-f002:**
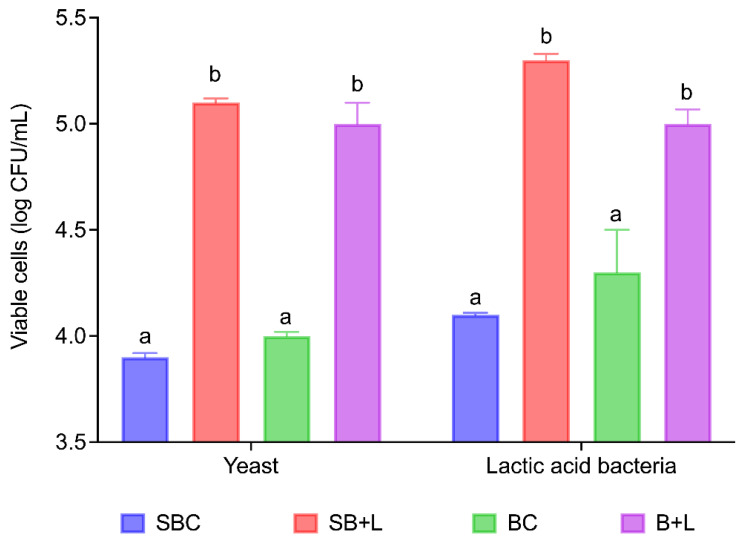
Microbial cell density at the end of bread fermentation. Different letters denote statistically significant differences (*p* < 0.05) between different formulations of bread. SBC: control sourdough bread; SB+L: sourdough with 5% Cava lees; BC: control bread; B+L: bread with 5% Cava lees.

**Figure 3 foods-11-01361-f003:**
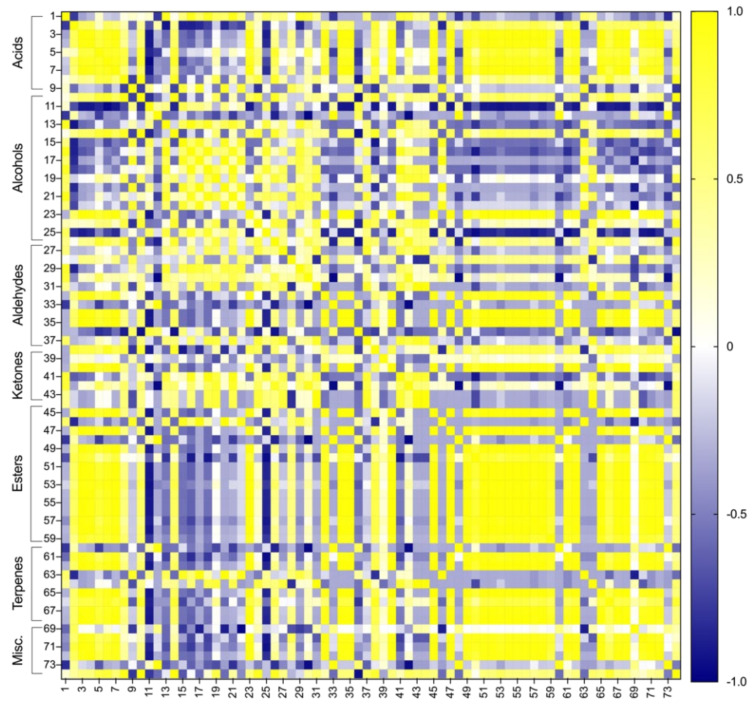
Heatmap of the correlation matrix of the volatile compounds (*p* < 0.05). Numbers correspond to the volatile compounds identified in Table 4. Positive correlations are shown in yellow; negative correlations in blue; absence of correlation in white.

**Figure 4 foods-11-01361-f004:**
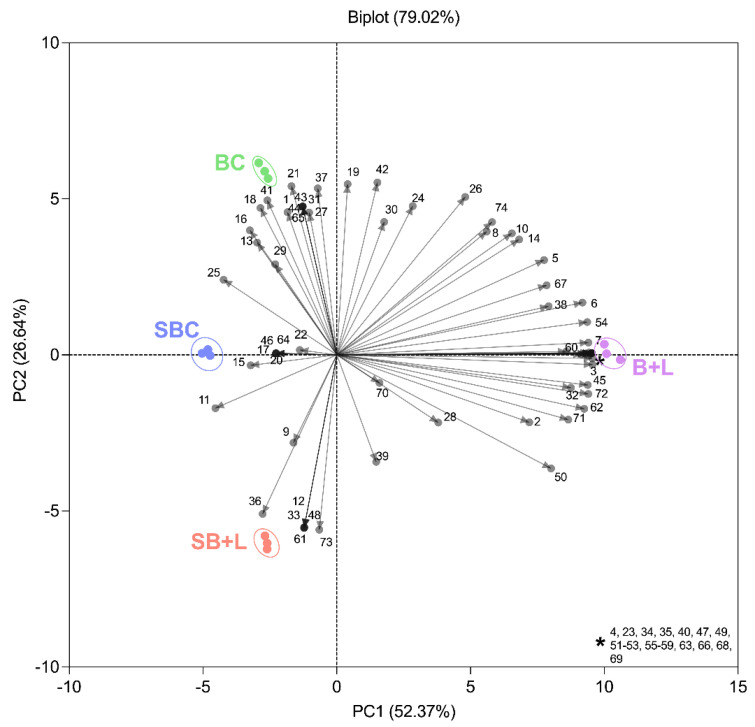
Principal Component Analysis (PCA) biplot of the breads obtained. SBC: control sourdough bread; SB+L: sourdough with 5% Cava lees; BC: control bread; B+L: bread with 5% Cava lees. Numbers correspond to the volatile compounds identified in Table 4.

**Table 1 foods-11-01361-t001:** Ingredients of sourdough (wheat flour weight basis, g).

Code	Wheat Flour	Water	Dough ^1^	Cava Lees
SDC	100	100	100	-
SD+L	100	100	100	5 ^2^

^1^ Aliquot of the previous dough into de new mixture. ^2^ Lees were added as a percentage of flour weigh in sourdough formulation in each propagation step.

**Table 2 foods-11-01361-t002:** Ingredients of bread (wheat flour weight basis, g).

Code ^1^	Wheat Flour	Water	Sourdough	Baker’s Yeast	Salt	Cava Lees
SBC	500	285	150	4	10	-
SB+L	500	285	150	4	10	-
BC	500	285	-	4	10	-
B+L	500	285	-	4	10	25

^1^ Codes of sample series of bread types: SBC: control sourdough bread; SB+L: sourdough bread with 5% Cava lees; BC: control bread; B+L: bread with 5% Cava lees.

**Table 3 foods-11-01361-t003:** pH values at the beginning (t = 0 h) and end (t = 2 h) of bread fermentation.

pH	SBC	SB+L	BC	B+L
t = 0 h	5.48 ± 0.03 ^a^	5.00 ± 0.01 ^b^	5.76 ± 0.03 ^c^	4.97 ± 0.03 ^b^
t = 2 h	5.14 ± 0.02 ^a^	4.75 ± 0.04 ^b^	5.45 ± 0.02 ^c^	4.37 ± 0.03 ^d^

Values are mean ± standard deviation of triplicates. Significant differences between samples are indicated by different superscript letters (*p* < 0.05) for each compound. SBC: control sourdough bread; SB+L: sourdough bread with 5% Cava lees; BC: control bread; B+L: bread with 5% Cava lees.

**Table 4 foods-11-01361-t004:** Concentration (mg/kg) of the main volatile compounds identified in bread.

	Compound	CAS-Num.	Odor ^1^	ODT ^2^	SBC	SB+L	BC	B+L
	**ACIDS**							
1	Acetic acid	64-19-7	sharp pungent sour vinegar	-	143.83 ± 1.63 ^a^	27.01 ± 8.74 ^b^	132.04 ± 0.06 ^a^	67.60 ± 16.22 ^c^
2	Benzoic acid	65-85-0	faint balsam urine	na	nd	6.83 ± 0.98 ^a^	3.89 ± 0.10 ^a^	8.52 ± 2.35 ^a^
3	Decanoic acid	334-48-5	sweet waxy floral soapy clean	1000	nd	5.09 ± 0.77 ^a^	nd	77.19 ± 2.62 ^b^
4	Dodecanoic acid	143-07-7	sweet waxy floral soapy clean	1000	nd	nd	nd	240.41 ± 83.22
5	Hexadecanoic acid	57-10-3	slightly waxy fatty	1000	7.26 ± 1.78 ^a^	nd	10.20 ± 1.03 ^a^	18.86 ± 2.66 ^b^
6	Hexanoic acid	142-62-1	sour fatty sweat cheese	300	nd	3.76 ± 0.49 ^a^	18.33 ± 0.33 ^b^	46.97 ± 5.34 ^c^
7	Octanoic acid	124-07-2	fatty waxy rancid oily vegetable cheesy	300	13.69 ± 2.56 ^a^	nd	11.13 ± 6.27 ^a^	173.53 ± 11.96 ^b^
8	Isobutyric acid	79-31-2	acidic sour cheese dairy buttery rancid	810	nd	nd	18.90 ± 6.30 ^a^	18.72 ± 9.86 ^a^
9	Myristic acid	544-63-8	waxy fatty soapy coconut	1000	28.54 ± 3.87 ^a^	12.51 ± 2.45 ^b^	nd	8.66 ± 1.53 ^b^
	TOTAL ACIDS				193.32 ± 9.84 ^a^	55.20 ± 13.43 ^a^	194.49 ± 14.09 ^a^	660.46 ± 135.76 ^b^
	**ALCOHOLS**							
10	Butyl alcohol	71-36-3	fusel oil sweet balsam whiskey	50	nd	nd	10.32 ± 1.97 ^a^	11.61 ± 2.65 ^a^
11	Isoamyl alcohol	123-51-3	fusel oil alcoholic whiskey fruity banana	25–30	96.16 ± 7.30 ^a^	92.88 ± 6.19 ^a^	82.69 ± 4.57 ^a^	60.26 ± 5.17 ^b^
12	1-Dodecanol	112-53-8	earthy soapy waxy fatty honey coconut	na	nd	4.76 ± 0.30	nd	nd
13	1-Hexanol	111-27-3	ethereal fusel oil fruity alcoholic sweet green	250	88.78 ± 31.46 ^a^	37.95 ± 11.30 ^b^	76.65 ± 11.19 ^ab^	39.54 ± 7.10 ^b^
14	2-Ethyl-1-hexanol	104-76-7	citrus fresh floral oily sweet	na	8.62 ± 3.20 ^a^	9.81 ± 4.44 ^a^	55.69 ± 9.96 ^b^	64.75 ± 1.08 ^b^
15	1-Octanol	111-87-5	waxy green orange aldehydic rose mushroom	11–13	18.61 ± 8.18 ^a^	11.22 ± 2.88 ^a^	10.23 ± 2.45 ^a^	6.92 ± 2.57 ^a^
16	1-Octen-3-ol	3391-86-4	mushroom earthy green oily fungal raw chicken	1	20.22 ± 2.36 ^a^	8.89 ± 0.87 ^b^	20.17 ± 1.25 ^a^	8.06 ± 0.96 ^b^
17	1-Pentanol	71-41-0	fusel oil sweet balsam	400	13.85 ± 1.48	nd	nd	nd
18	2-Methyl-1-propanol	78-83-1	ethereal winey cortex	na	8.79 ± 0.79 ^a^	nd	12.27 ± 0.84 ^b^	nd
19	2-Furanmethanol	98-00-0	alcoholic chemical musty sweet caramel bread coffee	na	13.79 ± 5.67 ^a^	7.40 ± 2.44 ^a^	24.12 ± 4.04 ^b^	15.54 ± 1.78 ^ab^
20	7-Octen-4-ol	53907-72-5	-	na	28.42 ± 5.87	nd	nd	nd
21	9-Decen-1-ol	13019-22-2	dewy rose waxy fresh clean aldehydic	na	27.08 ± 3.27 ^a^	7.78 ± 0.86 ^b^	34.13 ± 5.97 ^a^	15.69 ± 3.48 ^b^
22	Phenethyl alcohol	60-12-8	floral rose dried rose flower rose water	75–110	253.45 ± 3.78 ^a^	100.51 ± 33.46 ^b^	102.35 ± 17.03 ^b^	130.47 ± 30.62 ^b^
23	Benzyl alcohol	100-51-6	floral rose phenolic balsamic	1000	nd	nd	nd	11.64 ± 1.27
24	2-Phenoxy-ethanol	122-99-6	mild rose balsam cinnamyl	na	nd	nd	15.46 ± 5.11 ^a^	8.76 ± 2.48 ^a^
25	Heptanol	111-70-6	musty leafy violet herbal green sweet woody peony	0.3	22.69 ± 5.53 ^a^	12.77 ± 1.97 ^b^	24.61 ± 2.15 ^a^	nd
26	Nonanol	143-08-8	fresh clean fatty floral rose orange dusty wet oily	5	10.81 ± 1.48 ^a^	nd	19.48 ± 6.17 ^b^	21.54 ± 1.57 ^b^
	TOTAL ALCOHOLS				611.27 ± 80.37 ^a^	293.97 ± 64.71 ^b^	488.17 ± 72.70 ^ac^	394.78 ± 60.73 ^bc^
	**ALDEHYDES**							
27	(E)-2-Heptenal	18829-55-5	pungent green vegetable fresh fatty	1.3	16.50 ± 2.36 ^a^	7.58 ± 3.29 ^a^	40.93 ± 20.65 ^ab^	13.21 ± 3.08 ^a^
28	(E)-2-Nonenal	18829-56-6	fatty green cucumber aldehydic citrus	0.08-0.1	27.29 ± 2.93 ^a^	10.85 ± 0.93 ^b^	nd	26.21 ± 3.28 ^a^
29	(E)-2-Octenal	2548-87-0	fresh cucumber fatty green herbal banana waxy green leaf	0.3	23.02 ± 2.44 ^a^	10.77 ± 3.39 ^b^	17.74 ± 1.86 ^ab^	12.81 ± 1.84 ^b^
30	(E,E)-2,4-Decadienal	25152-84-5	oily cucumber melon citrus pumpkin nut meat	0.07	10.46 ± 2.36 ^a^	nd	8.23 ± 0.94 ^a^	8.92 ± 0.64 ^a^
31	(E,Z)-2,4-Decadienal	25152-83-4	fried fatty geranium green waxy	na	nd	nd	24.70 ± 2.79	nd
32	Benzaldehyde	100-52-7	strong sharp sweet bitter almond cherry	35–350	41.90 ± 4.59 ^a^	43.46 ± 19.19 ^a^	24.36 ± 5.68 ^a^	88.66 ± 11.41 ^b^
33	*o*-Tolualdehyde	529-20-4	cherry	na	nd	4.57 ± 0.57	nd	nd
34	Butanal	123-72-8	pungent cocoa musty green malty bready	0.9–3.73	nd	nd	nd	2.39 ± 0.15
35	Isovaleraldehyde	590-86-3	ethereal aldehydic chocolate peach fatty	0.2–2	nd	nd	nd	2.05 ± 0.43
36	Heptanal	111-71-7	fresh aldehydic fatty green herbal wine-lee ozone	3	14.04 ± 7.68 ^a^	18.19 ± 1.69 ^a^	nd	nd
37	Hexanal	66-25-1	fresh green fatty aldehydic grass leafy fruity sweaty	4.5–5	56.11 ± 11.12 ^a^	32.86 ± 14.76 ^a^	105.01 ± 16.26 ^b^	53.56 ± 11.53 ^a^
38	Nonanal	124-19-6	waxy aldehydic rose fresh orris orange peel fatty peely	1	nd	8.88 ± 0.81 ^a^	15.04 ± 0.74 ^b^	21.80 ± 2.63 ^c^
	TOTAL ALDEHYDES				189.68 ± 33.48 ^a^	137.16 ± 44.63 ^a^	236.01 ± 48.92 ^a^	229.61 ± 34.99 ^a^
	**KETONES**							
39	Acetoin	513-86-0	sweet buttery creamy dairy milky fatty	80	13.19 ± 6.75 ^a^	14.03 ± 9.11 ^a^	6.94 ± 0.97 ^a^	13.38 ± 3.59 ^a^
40	2-Nonanone	821-55-6	fresh sweet green weedy earthy herbal	0.5–20	nd	nd	nd	4.78 ± 1.28
41	2-Octanone	111-13-7	earthy weedy natural woody herbal	5	2.64 ± 0.45 ^a^	nd	4.87 ± 1.04 ^b^	nd
42	4-Methyl-2-pentanone	108-10-1	sharp solvent green herbal fruity dairy spice	na	11.24 ± 3.49 ^a^	nd	14.52 ± 0.78 ^a^	11.26 ± 2.37 ^a^
43	2,3-Octanedione	585-25-1	dill asparagus cilantro herbal aldehydic earthy fatty cortex	na	nd	nd	6.26 ± 1.08	nd
44	Acetophenone	98-86-2	sweet pungent hawthorn mimosa almond acacia chemical	6.5	nd	nd	5.64 ± 1.11	nd
	TOTAL KETONES				27.07 ± 10.69 ^ab^	14.03 ± 9.11 ^b^	38.23 ± 4.98 ^a^	29.42 ± 7.24 ^ab^
	**ESTERS**							
45	Isoamyl decanoate	2306-91-4	waxy banana fruity sweet cognac green	na	nd	5.54 ± 0.83 ^a^	nd	175.37 ± 20.93 ^b^
46	Phenethyl acetate	103-45-7	floral rose sweet honey fruity tropical	na	6.20 ± 1.12	nd	nd	nd
47	Hexyl acetate	142-92-7	fruity green apple banana sweet	2	nd	nd	nd	12.77 ± 2.57
48	L-Bornyl acetate	5655-61-8	sweet balsamic woody fresh pine needle herbal	na	nd	6.23 ± 0.98	nd	nd
49	Diethyl succinate	123-25-1	mild fruity cooked apple ylang	na	nd	nd	nd	47.49 ± 6.39
50	Ethyl decanoate	628-97-7	mild waxy fruity creamy milky balsamic greasy oily	>200	59.56 ± 3.91 ^a^	127.37 ± 13.90 ^b^	nd	220.30 ± 38.41 ^b^
51	Ethyl 9-hexadecenoate	54546-22-4	-	na	nd	nd	nd	23.95 ± 2.64
52	Ethyl 9-decenoate	67233-91-4	fruity fatty	na	nd	nd	nd	41.81 ± 2.93
53	Ethyl hexanoate	123-66-0	sweet fruity pineapple waxy green banana	1	9.67 ± 3.37 ^a^	nd	18.64 ± 3.78 ^b^	98.44 ± 3.07 ^c^
54	Octyl acetate	112-14-1	green earthy mushroom herbal waxy	1.2	nd	nd	nd	8.04 ± 1.26
55	Ethyl nonadecanoate	18281-04-4	-	na	nd	nd	nd	5.43 ± 1.02
56	Isoamyl octanoate	2035-99-6	sweet oily fruity green soapy pineapple coconut	na	nd	nd	nd	181.00 ± 17.83
57	Ethyl octanoate	106-32-1	fruity wine waxy sweet apricot banana brandy pear	na	nd	nd	28.72 ± 4.55 ^a^	965.31 ± 167.18 ^b^
58	Phenethyl isobutyrate	103-48-0	floral fruity rose tea rose peach pastry	na	nd	nd	nd	14.22 ± 2.49
59	Ethyl myristate	124-06-1	sweet waxy violet orris	na	nd	nd	nd	10.75 ± 7.36
	TOTAL ESTERS				75.43 ± 8.76 ^a^	139.14 ± 15.71 ^a^	47.36 ± 8.33 ^a^	1804.88 ± 274.08 ^b^
	**TERPENES**							
60	𝛼-Terpinolene	586-62-9	fresh woody sweet pine citrus	20	nd	8.71 ± 1.25	nd	nd
61	Vitispirane	66965-94-4	floral fruity earthy woody	na	nd	7.73 ± 0.43 ^a^	nd	25.03 ± 3.08 ^b^
62	(E,E)-Farnesyl acetate	4128-17-0	oily waxy	na	nd	nd	nd	44.88 ± 5.60
63	dihydromyrcenol	18479-58-8	fresh citrus lime floral clean	na	2.32 ± 0.13	nd	nd	nd
64	Bornylene	464-17-5	-	na	nd	nd	37.11 ± 6.92	nd
65	d-Nerolidol	142-50-7	mild floral	na	nd	nd	nd	13.66 ± 2.01
66	DL-Limonene	138-86-3	citrus herbal terpene camphor	10	5.39 ± 1.06 ^a^	9.43 ± 4.22 ^a^	23.34 ± 14.40 ^ab^	34.02 ± 2.30 ^b^
67	Farnesol	4602-84-0	mild fresh sweet linden floral angelica	2	nd	nd	nd	24.84 ± 6.21
68	Nerolidol	7212-44-4	floral green waxy citrus woody	na	nd	nd	nd	73.31 ± 8.75
	TOTAL TERPENES				7.71 ± 1.19 ^a^	25.87 ± 5.90 ^ab^	60.45 ± 21.32 ^b^	215.74 ± 27.95 ^c^
	**MISCELANEOUS**							
69	2-Pentyl- furan	3777-69-3	fruity green earthy beany vegetable metallic	6	22.82 ± 1.11 ^a^	59.05 ± 9.20 ^b^	54.47 ± 6.62 ^b^	46.31 ± 6.37 ^b^
70	2-Furancarboxaldehyde	98-01-1	sweet woody almond fragrant baked bread	na	6.20 ± 3.68 ^a^	6.45 ± 0.91 ^a^	nd	19.55 ± 8.56 ^b^
71	1,1,6-Trimethyl-1,2-dihydronaphthalene (TDN)	30364-38-6	licorice	na	nd	43.95 ± 16.26 ^a^	nd	196.29 ± 34.35 ^b^
72	4-Ethylguaiacol	2785-89-9	spicy smoky bacon phenolic clove	50	nd	nd	nd	19.73 ± 6.59
73	Styrene	100-42-5	sweet balsam floral plastic	730	nd	103.69 ± 30.91 ^a^	nd	12.59 ± 0.89 ^b^
74	𝜸-Nonalactone	104-61-0	coconut creamy waxy sweet buttery oily	na	nd	nd	15.80 ± 7.79 ^a^	15.10 ± 4.86 ^a^
	TOTAL MISCELANEOUS				29.02 ± 4.79 ^a^	213.14 ± 57.28 ^b^	70.27 ± 14.41 ^a^	309.56 ± 61.62 ^b^

^1^ From [27]. ^2^ ODT: Odor Detection Threshold. From [28]. Expressed as µg/mL. Values are mean ± standard deviation of triplicates. Significant differences between samples are indicated by different superscript letters (*p* < 0.05) for each compound. na: not available; nd: not detected. SBC: control sourdough bread; SB+L: sourdough bread with 5% Cava lees; BC: control bread; B+L: bread with 5% Cava lees.

## Data Availability

The data presented in this study are available in this article and Appendix A.

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
