# Peer review of "By-Product Revalorization: Cava Lees Can Improve the Fermentation Process and Change the Volatile Profile of Bread"

_foods, 2022, doi:10.3390/foods11091361_

Round 1

Reviewer 1 Report

The manuscript entitled „By-Product Revalorization: Cava Lees Improve the Fermentation Process and Sensory Profile of Bread” describes a study in which the volatile organic compounds were analysed in bread obtained after fortification with Cava lees.

It is rather a confusing study. The selected by-product did not show any additional, functional value (which was not studied here). What is the point of fortification, if there is no benefit for the product? There was no improvement in neither sensory properties (which were not analysed) nor bioactive potential or health beneficial properties (which were also not analysed). Analysis of VOCs suggests that the authors wanted to improve the aroma of the product, but it was not analysed and confirmed. The article needs sensory analysis.

If the aroma of the product was the goal, then SAFE instead of SPME (a competitive method) should be used. Then the key odourants could be defined.

Also, it is so strange that VOCs were analysed in fortified bread but not in the raw material (Cava lees). How can it be compared and how the authors can guess that some compounds were derived from by-products if it was not analysed in this study.

It would be expected that bread obtained from sourdough has a more rich aroma profile. In this study, the sourdough bread fortified with lees had the smallest abundance of main aroma VOCs. How is it possible? It should be explained in discussion, not omitted.

Minor issues:

Line 128: Agilent detector of Thermo? It is something strange here.

Section 2.4. was it manual injection or with an autosampler?

Did the authors calculate the linear indices?

If only the library was used for confirmation of compounds, what were the selection criteria? And then it should be called “tentative identification”

Line 154: the first sentence is misleading. It suggests that two types of sourdough were made. Which are 2 separate experimental settings. But in this study, there was one type of sourdough, just fortified, resulting in the experimental and control sample. It is not two experiments.

There is a problem with the origin of VOCs. It is not as easy as described (lines 196-204). E.g. alcohols also can originate from lipid oxidation. This should be rewritten.

Line 275: not all terpenes have flower aroma.

Author Response

Reviewer 1:

Thank you for your comments and suggestions in order to improve the paper.

The selected by-product, yeast lees were selected due to its potential effect in fermentation foods. In this sense, on 2011 (Decision 2011/762) the b-glucans of yeast were authorized as a new ingredient for food industry for a variety of foods. Also, it could be found in bibliography as other industry by-products and co-products as broccoli stem and leaves, cumin and caraway seeds and cocoa dietary fiber were added to sourdough or bread in order to obtain functional products or modify the sensory properties of the final bread. Furthermore, the previous papers and experience of the research group showed that the yeast lees could improve the survival and growth of LAB and yeast and, at the same time, they inhibit the growth of pathogenic bacteria such as Salmonella spp. and L. Monocytogenes

Regarding the need to include sensory analysis in the present paper, we agree that is necessary to confirm the results obtained by chromatographic analysis of VOC. In this sense, we are working in a complete sensory analysis in order to describe the resulting breads. This requires time for the selection and training of the panelists and actually we are working on it. Regarding the extraction method used by the VOC determination, we are agreeing than SAFE could be a good method in order to obtain information about the key odorants of the samples instead of SPME. However, SPME coupled to GC-MS has provided to be adequate enough in order to obtain a wide volatile profile and it is useful to compare samples, when the same extraction conditions were used.

VOC were also, analyzed in raw material, yeast lees. The information about VOC were added in the paper as Supplementary material. Results were obtained with the same method used by the bread.

According to the previous bibliography sourdough modifies the volatile composition of the final product. This depends of multiple factors as the fermentative microorganisms, the age of sourdough, the type of sourdough, etc. therefore, predicting the volatile profile is very complicated. A possible explanation was added in the results and discussion section (pages 226-230).

Minor issues were corrected.

Line 138: the detector used was Agilent, it was corrected in the text.

Section 2.4: the samples were manually injected. It was added to the text (line 142).

Linear retention indices were not calculated. The commercial library was the main method used to identify volatile compounds as well as the information of references. However, tentative identification was added in the text.

Line 154: the sentence was corrected in order to clarify this point (Line 164).

Lines 196-204: The origin of volatile compounds in bread could be from fermentation, ingredients, and the reactions that take place during bread production: Maillard, caramelization, or lipid oxidation. According to the literature consulted the main pathways are that cited in the text but it’s correct that is more complex. It’s true that the chemical families of VOC could be formed by different pathways, for example alcohols could provide from lipid oxidation and at also from fermentation. However, the paragraph was rewritten in order to clarify.

Line 275: it’s true that not all the terpenes are flowery. However, mostly of them provide floral odor. It was clarified in the text. It could be observed the odor of terpenes identified in Table 4. 

Reviewer 2 Report

The submitted article evaluated the effect of Cava lees on microbiological populations during sourdough and bread fermentation and the volatile fraction of the final bread. I find this paper to be an interesting and relevant topic as the topic of by-product revalorization is very important and popular at present. The presented idea has potential; however, the experimental plan is modest. For example, how did the authors come to 5% of Cava lees? Is it a result of optimization? These results should be included in the article.
Page 1, Include names and title of the article in the citation;
The flow of background information is not appropriate;
Page 3, 97 lines, why was the dough manually mixed?
Please, provide below the tables, the meaning of the different letters in Table 3;
Page 11, please, Include correlation analysis before the Principal Component Analysis biplot. Perhaps a color correlation diagram?
Page 11, please, elaborate on variable contributions obtained by Principal Component Analysis;
I recommend thorough editing of the paper; please correct spelling and grammatical issues.

Author Response

Reviewer 2:

Thank you for your comments and suggestions in order to improve the paper.

The 5% of Cava lees used is due to the previous experience. Different concentrations of yeast lees have been tested with In Vitro studies and 5% of lees were the best one for the growth and survival of microorganisms. Later, this 5% was used with satisfactory results in fermented sausages. It was explained in Material and Methods (Lines 92-94).

Page 1: the name and title were added in the citation.

Also, more information was added in the introduction in order to clarify the information.

Page 3, line 67: the dough was manually mixed because the breads and sourdoughs were made at a laboratory scale. Although, in the future and for more volume, it could be interesting to use a mixer.

Table 3: the meaning of the different letters were added.

Page 11: According to the suggestion of the reviewer a Heat map of correlation matrix was added previously to the PCA. Also, the PCA was better explained in the text (Lines 315-325).

Reviewer 3 Report

Utilization of winery by-product has high relevance for the practice. Authors deal with the effect of Cava lees on sourdough and bread fermentation, and the concentration of volatiles. Therefore, the topic of the manuscript is interesting and relevant. The manuscript is generally well structured. Introduction section summarizes well the relevance and the background of the study. Apllied methods (microbial monitoring, analytical methods) are adequate. Materials and methods are described clearly and in details. The manuscript contains interesting results that are valuable not just for the science, but also for the practice. Results are discussed in details, with relevant references.

Comments, suggestions:

Please give the acronyms in the first place int he manuscript (see line 17, LAB).

It is not clear how the maximum concentration of cava lees was selected/determined.

The visibility of y axis titles in Figure 1 is low, please improve them.

I suggest to use 2 or 3 separated table instead of Table 4.

Author Response

Reviewer 3:

Thank you for your comments and suggestions in order to improve the paper.

Acronyms have been indicated in the first place.

The 5% of Cava lees used is due to the previous experience. Different yeast lees have been tested with In Vitro studies and 5% of lees were the best one for the growth and survival of microorganisms. Later, this 5% was used with satisfactory results in fermented sausages. It was explained in Material and Methods (Lines 92-94).

Figure 1 has been posted again with higher quality.

Table 4 is too long, we have decided to leave it the same to make it easier to compare the different total concentrations.

Round 2

Reviewer 1 Report

I am satisfied with the answers

Author Response

Thank you for your comments!

Reviewer 2 Report

The authors' answers are fair. The article is improved.

Author Response

Thank you for your comments!